# Epidermolysis-Bullosa-Associated Squamous Cell Carcinomas Support an Immunosuppressive Tumor Microenvironment: Prospects for Immunotherapy

**DOI:** 10.3390/cancers16020471

**Published:** 2024-01-22

**Authors:** David Rafei-Shamsabadi, Lena Scholten, Sisi Lu, Daniele Castiglia, Giovanna Zambruno, Andreas Volz, Andreas Arnold, Mina Saleva, Ludovic Martin, Kristin Technau-Hafsi, Frank Meiss, Dagmar von Bubnoff, Cristina Has

**Affiliations:** 1Department of Dermatology, Medical Center—University of Freiburg, Faculty of Medicine, 79104 Freiburg, Germany; lena.scholten@diakoneo.de (L.S.); lusisi_1984@hotmail.com (S.L.); kristin.technau@uniklinik-freiburg.de (K.T.-H.); frank.meiss@uniklinik-freiburg.de (F.M.); cristina.has@uniklinik-freiburg.de (C.H.); 2Department of Obstetrics and Gynaecology, Union Hospital, Tongji Medical College, Huazhong University of Science and Technology, Wuhan 430022, China; 3Laboratory of Molecular and Cell Biology, Istituto Dermopatico dell’Immacolata Istituto di Ricovero e Cura a Carattere Scientifico (IDI-IRCCS), Via Monti di Creta 104, 00167 Rome, Italy; d.castiglia@idi.it; 4Genetics and Rare Diseases Research Division, Bambino Gesù Children’s Hospital, Istituto di Ricovero e Cura a Carattere Scientifico (IRCCS), 00165 Rome, Italy; giovanna.zambruno@opbg.net; 5Dermatologie am Rhein, 4051 Basel, Switzerlandandreas.arnold@hin.ch (A.A.); 6Department of Dermatology and Venereology, University Hospital “Alexandrovska”, Faculty of Medicine, Sofia University of Medicine, 1431 Sofia, Bulgaria; m.saleva@medfac.mu-sofia.bg; 7MAGEC Nord Reference Center for Rare Skin Diseases, Department of Dermatology, Angers University Hospital, 49933 Angers, France; lumartin@chu-angers.fr; 8Department of Dermatology, Allergology and Venerology, University Hospital Schleswig-Holstein, Campus Lübeck, 23538 Lübeck, Germany; dagmarangela.vonbubnoff@uksh.de

**Keywords:** epidermolysis bullosa, collagen VII, kindlin, squamous cell carcinoma, indoleamine 2,3-dioxygenase, programmed cell death protein-1, programmed cell death ligand-1

## Abstract

**Simple Summary:**

Cutaneous squamous cell carcinomas (SCCs) are the most common tumors in patients suffering from the genetic disorder epidermolysis bullosa (EB). These tumors frequently show aggressive growth, leading to metastatic disease and fatal outcomes. Therapeutic options in advanced stages are limited, and new drug targets are needed. The present study evaluated the immune microenvironment of cutaneous SCCs, which is assumed to favor local immunosuppression and lead to a more severe disease course in patients. The expression of several immune checkpoint molecules in tumor cells and cells in the tumor microenvironment was evaluated in EB-SCCs and compared with those in SCCs from immunocompetent and immunosuppressed patients. Our results show high expression of the immunosuppressive markers indoleamine 2,3-dioxygenase (IDO), PD-1, and programmed cell death ligand-1 (PD-L1) in tumor cells from dystrophic EB (DEB) patients.

**Abstract:**

Cutaneous squamous cell carcinomas (SCCs) are a major complication of some subtypes of epidermolysis bullosa (EB), with high morbidity and mortality rates and unmet therapeutic needs. The high rate of endogenous mutations and the fibrotic stroma are considered to contribute to the pathogenesis. Patients with dystrophic EB (DEB) and Kindler EB (KEB) have the highest propensity for developing SCCs. Another patient group that develops high-risk SCCs is immunosuppressed (IS) patients, especially after organ transplantation. Herein, we interrogate whether immune checkpoint proteins and immunosuppressive enzymes are dysregulated in EB-associated SCCs as an immune resistance mechanism and compare the expression patterns with those in SCCs from IS patients, who frequently develop high-risk tumors and sporadic SCCs, and immunocompetent (IC) individuals. The expression of indoleamine 2,3-dioxygenase (IDO), programmed cell death protein-1 (PD-1), programmed cell death ligand-1 (PD-L1), T cell immunoglobulin and mucin-domain-containing protein-3 (TIM-3), lymphocyte activation gene-3 (LAG-3), and inflammatory infiltrates (CD4, CD8, and CD68) was assessed via immunohistochemistry and semi-quantitative analysis in 30 DEB-SCCs, 22 KEB-SCCs, 106 IS-SCCs, and 100 sporadic IC-SCCs. DEB-SCCs expressed significantly higher levels of IDO and PD-L1 in tumor cells and PD-1 in the tumor microenvironment (TME) compared with SCCs from IC and IS individuals. The number of CD4-positive T cells per mm^2^ was significantly lower in DEB-SCCs compared with IC-SCCs. KEB-SCCs showed the lowest expression of the exhaustion markers TIM-3 and LAG-3 compared with all other groups. These findings identify IDO, PD-1, and PD-L1 to be increased in EB-SCCs and candidate targets for combinatory treatments, especially in DEB-SCCs.

## 1. Introduction

Specific subtypes of epidermolysis bullosa (EB) are characterized by an increased risk of squamous cell carcinomas (SCCs) that arise in cutaneous and mucosal sites of chronic tissue damage [1,2,3]. Dystrophic EB (DEB) and Kindler EB (KEB) are associated with multiple SCCs that occur at a young age, metastasize early, and have a severe, lethal course [4,5,6,7]. In DEB, SCCs were reported at as early as 13 years of age, with high mortality [8]. In KEB, SCCs arise in sites with high UV exposure or mechanical stress [4,6,9]. The youngest patient reported with a KEB-associated SCC (KEB-SCC) was 6 years old [10]. Currently, wide excision and amputation are the standard treatments [4]. Information about the effectiveness of immunotherapy for advanced EB-SCC is based on case reports and small case series [4,11,12,13,14,15,16,17,18,19]. There is a high unmet therapeutic need for these patients.

In the skin of DEB patients, endogenous mutation processes dominated by apolipoprotein B mRNA-editing enzyme, catalytic polypeptide-like (APOBEC) take place and lead to a high number of early-acquired mutations [20]. Cellular stress and inflammation resulting from continuous tissue damage and microbial insult are sources of APOBEC activation [20]. No information on the mutational spectrum of KEB-SCC is available. Nevertheless, kindlin-1 is involved in regulating cell cycle arrest and DNA damage [21] and different types of cancer [22,23,24,25].

Fibroblast activation, transforming growth factor beta-1, and other pathways involved in tissue repair are common features of DEB and KEB [26,27,28,29,30,31,32]. The fibrotic stroma is considered to have a determinant role in the aggressive course of DEB-SCC. The mechanisms of inflammation in EB are multi-layered, involving tissue repair and bacterial colonization, both contributing to carcinogenesis [33]. Recent in vitro data suggest that adaptive T-cell-mediated immunity might be inhibited by PD-1/Treg-mediated immunosuppression in DEB [34]. Nevertheless, little is known about the immunological microenvironment of EB-SCC.

The upregulation of immune checkpoint markers on cells of the tumor and tumor microenvironment (TME) can profoundly shape the antitumor response of the organism. Immune checkpoint molecules, such as programmed cell death protein-1 (PD-1) on T cells and programmed cell death ligand-1 (PD-L1) on antigen-presenting or cancer cells, play an important role in tumor surveillance, and targeting these molecules with monoclonal antibodies leads to increased T cell activation [35,36]. The enzyme indoleamine 2,3-dioxygenase (IDO), which degrades L-tryptophan to kynurenine, supports an immunosuppressive tumor environment by promoting Treg differentiation [37,38]. T cell exhaustion markers, such as T cell immunoglobulin and mucin-domain-containing protein-3 (TIM-3) and lymphocyte activation gene-3 (LAG-3), can also be targeted with antibodies to reduce T cell dysfunction and increase the number of activated tumor-killing T cells [39,40].

This study aimed to investigate whether EB-SCCs express immune checkpoint markers that may contribute to the severe course of the disease and be therapeutically targeted.

## 2. Materials and Methods

### 2.1. Patients and Samples

A total of 258 cutaneous SCCs were studied; 30 SCCs were from recessive DEB patients (n = 9), 22 SCCs were from KEB patients (n = 7), 100 were sporadic SCCs on the sun-exposed skin of immunocompetent patients (IC-SCCs) (n = 96), and 106 were SCCs excised from immunosuppressed patients (IS-SCCs) (n = 41) (Table 1). The EB patients did not receive anti-proliferative drugs, immunotherapy, or radiotherapy before the excision of the tumors, except for DEB-P1, who received anti-PD-1 therapy before the tumor in the right hand was removed [15] (Appendix A). Immunosuppressed patients had one of the following conditions: organ transplantation, stem cell therapy, inflammatory autoimmune disease, congenital immune deficiency, solid tumors, or hematological malignancy (Appendix A). The immunosuppressive medication is included in Appendix A. All tumors were excised for therapeutic purposes. The histopathological characteristics of the tumors were retrieved from the pathology reports, and the original hematoxylin and eosin (H&E)-stained slides were reviewed. This study was approved by the Ethical Committee of the University of Freiburg (EK-Freiburg 45/18 and 5/20) and conducted according to the Principles of Helsinki.

### 2.2. Immunohistochemistry

Serial sections of 5 μm were prepared from formalin-fixed, paraffin-embedded skin biopsies. Standard hematoxylin and eosin staining was performed for diagnostic purposes. For immunohistochemical (IHC) staining, sections were deparaffinized and subjected to heat-induced antigen retrieval using EDTA (pH 9) or citrate (pH 6) retrieval buffer. For some stainings, sections underwent a block of non-specific staining via incubation with a blocking buffer (2% BSA + 0.05% Tween 20 in TBS) for 30 min at room temperature. The primary antibodies used for incubation on deparaffinized tissue sections and the staining conditions are included in Appendix A. Primary antibodies were diluted in antibody diluent (Zytomed; catalog number: ZUC025). Incubation was performed at room temperature for 60 min. Sections incubated with secondary antibodies only served as staining controls. Visualization was performed using the Dako REAL™ detection system, alkaline phosphatase/RED, and rabbit/mouse (Dako, catalog number: K5005). Photographs were taken using a Zeiss microscope (Axioscope) (Carl Zeiss, Jena, Germany) and were visualized with the program AxioVision (Carl Zeiss, Jena, Germany, SE64 release 4.9).

Histology scores (H-scores) were calculated for the IHC markers IDO, PD-1, PD-L1, TIM-3, and LAG-3 as a semiquantitative approach using the following formula: H-score = [1 × (% cells 1+) + 2 × (% cells 2+) + 3 × (% cells 3+)]. Membrane staining intensity was assessed as 0 = no staining, 1+ = weak staining, 2+ = moderate staining, or 3+ = strong staining [8]. Three observers who were blinded to the clinical parameters scored the IHC-stained sections independently.

Absolute cell numbers per mm^2^ of CD4^+^, CD8^+^, and CD68^+^ cells were measured using the imaging software QuPath (Version 9.4.1) [41]. The average cell numbers of at least three different tumor sites (magnification ×10) were calculated using the positive cell detection and counting tool.

### 2.3. Statistical Analysis

The expression patterns were assessed in the SCC groups and compared with IC-SCCs. Vector graphics and statistical analysis were generated using GraphPad Prism version 9.4.1 for Windows (GraphPad Software, La Jolla, CA, USA, www.graphpad.com, accessed on 13 January 2024). Kruskal–Wallis tests and Dunn’s multiple comparison tests were used to compare the SCC groups. Simple linear regression was calculated to test for a correlation between vertical tumor thickness and H-scores. *p* < 0.05 was considered statistically significant.

## 3. Results

### 3.1. Clinicopathological Characteristics of SCCs

The clinicopathological characteristics of the studied patients and SCCs are included in Table 1, and the clinical and genetic findings for the EB patients are in Appendix A. Representative clinical pictures of SCCs from each subgroup are depicted in Figure 1. The mean age at SCC diagnosis was 32 years for DEB (range: 18–50), 47 years (range: 30–65) for KEB, 80 years (range: 45–99) for IC, and 68 years (range: 41–87) for IS patients. DEB- and KEB-SCCs were mainly localized in the upper and lower extremities (100% and 68%, respectively), while IC- and IS-SCCs were mainly distributed in the head and neck (79% and 51%, respectively) (Table 1). Half (50%) of the evaluated tumors showed a vertical tumor thickness of 2.01 to 6 mm. High-risk tumors with a vertical tumor thickness of >6 mm represented 17% of DEB-SCCs, 0% of KEB-SCCs, 18% of IC-SCCs, and 8% of IS-SCCs. In four DEB-SCCs and four IS-SCCs, the vertical tumor thickness could not be assessed due to partially fragmented specimens. The IS-SCC group included tumors from 34 organ transplant patients (22 kidney, 1 liver, 2 lung, 1 heart, and 8 stem cell transplants), 4 patients with inflammatory or autoimmune disease receiving immunosuppressive medication (two for rheumatoid arthritis, one for psoriasis, one for bullous pemphigoid), 1 patient with a congenital immune deficiency (CD4+ T cell defect), and 1 patient with B-CLL requiring treatment (Appendix A). The immunosuppressive medications comprised mTOR inhibitors (n = 5), calcineurin inhibitors (n = 24), purine synthesis inhibitors (n = 25), antimetabolites (n = 3), alkylating or alkylating-like agents (n = 2), and proteasome inhibitors (n = 1). Combination therapy of different immunosuppressive drugs was given in 23 cases (Appendix A).

### 3.2. DEB-SCC Tumor Cells Express Significantly Higher Levels of IDO and PD-L1 Compared with IC- and IS-SCC Tumor Cells

The expression of IDO in tumor cells was significantly higher in DEB-SCCs (mean H-score: 48.32) compared with KEB-SCCs (mean H-score: 10.8; *p* = 0.0038), IC-SCCs (mean H-score: 15.34; *p* < 0.0001), and IS-SCCs (mean H-score: 6.585; *p* < 0.0001) (Figure 2). No significant differences were found between IC- and IS-SCCs (mean H-scores: IC—15.34 vs. IS—6.585) (Figure 2). PD-L1 expression in DEB-SCCs was significantly higher compared with IC- and IS-SCCs (mean H-scores: DEB—96.59 vs. IC—26.36 vs. IS—33.26; *p* < 0.0001 and *p* = 0.0021, respectively). Similar results were obtained for KEB tumor cells (mean H-scores: KEB—62.39 vs. IC—26.36 vs. IS—33.26; *p* = 0.0015 and *p* = 0.0304, respectively). Notably, both IC-SCC and IS-SCC tumor cells demonstrated similar, extremely low IDO and PD-L1 expression (Figure 2).

Next, we analyzed the expression of IDO and PD-L1 in TME cells. IDO expression in the TME was generally high and comparable between most of the subgroups (DEB-SCC mean H-score: 108.1; IS-SCC mean H-score: 112.0; and IC-SCC mean H-score: 114.4). An exception was KEB-SCCs, which showed significantly lower IDO expression in the TME (KEB mean H-score: 46.14) (Figure 2). PD-L1 expression in the TME was comparable in all four groups (mean H-scores: DEB-SCC—75.73, IS-SCC—76.86, IC-SCC—73.57, and KEB—81.11) (Figure 2). Taken together, DEB-SCC tumor cells dramatically upregulate IDO and PD-L1 as markers of immune resistance that seem to play an important role in tumor development.

### 3.3. The TME Inflammatory Cell Infiltrate in EB-SCCs

Next, we quantified the inflammatory cells, CD4^+^ and CD8^+^ T cells, and CD68^+^ macrophages in the TME of 25–30% of the SCCs of each group (Figure 3). Not all samples could be assessed due to a shortage of tumor material, especially from EB patients. DEB-SCCs demonstrated significantly lower numbers of CD4^+^ T cells (mean: 649 cells/mm^2^) compared with IC-SCCs (mean: 1211 cells/mm^2^; *p* = 0.0039), and a similar trend was observed for IS-SCCs (mean: 1067 cells/mm^2^; *p* = 0.2084) (Figure 3). IS-SCCs showed the lowest number of CD8^+^ T cells (mean: 692/mm^2^) in the TME, which was significant compared with IC-SSCs (*p* = 0.0163). A similar trend was observed in DEB-SCCs (DEB mean—794/mm^2^ vs. IC—1228/mm^2^; *p* = 0.1501) (Figure 3). KEB-SCCs showed significantly lower numbers of CD68^+^ macrophages compared with IC-SCCs (KEB mean—338 cells/mm^2^ vs. IC mean—703 cells/mm^2^; *p* = 0.0113), but no other significant differences were observed. DEB-, IC-, and IS-SCCs showed similar CD68^+^ cell numbers (Figure 3). Altogether, the TMEs of DEB- and IS-SCCs demonstrated a reduced number of inflammatory infiltrates, especially tumor-infiltrating lymphocytes (TILs).

### 3.4. PD-1 Expression Was Significantly Upregulated in the TMEs of DEB-, KEB-, and IS-SCCs, While LAG-3 Was Increased in the TME of IS-SCCs

Next, we analyzed the T cell infiltrate in the TME in more detail and evaluated the immune checkpoint markers PD-1, TIM-3, and LAG-3 (Figure 4). PD-1 expression in the TME was significantly higher in DEB-SCCs (mean H-score: 151.8), KEB-SCCs (mean H-score: 132.1), and IS-SCCs (mean H-score: 118.6) compared with IC-SCCs (mean H-score: 85.34; *p* < 0.0001, *p* = 0.0439, and *p* = 0.0057, respectively). PD-1 expression was even higher in DEB-SCCs compared with IS-SCCs (mean H-score: 151.8 vs. 118.6; *p* = 0.1513), but this did not reach statistical significance (Figure 4a,b). TIM-3 expression was comparably high in IC-SCCs (mean H-score: 69.13), IS-SCCs (mean H-score: 75.01), and DEB-SCCs (mean H-score: 86.57) but significantly lower in the TME of KEB-SCCs (mean H-score: 25.93) (Figure 4a,b). LAG-3 expression was significantly higher in the TME of IS-SCCs (mean H-score: 53.07) compared with IC-SCCs (mean H-score: 31.68; *p* < 0.0001) and KEB-SCCs (mean H-score: 1.875; *p* < 0.0001) (Figure 4a,b). A trend, but no significant difference, was detected compared with DEB-SCCs (mean H-score 38.09; *p* = 0.1366). LAG-3 expression was significantly lower in KEB-SCCs compared with DEB-SCCs (mean H-scores 1.875 vs. 38.09; *p* = 0.0138). While PD-1 expression was increased in the TME of EB-SCCs, LAG-3 was upregulated in IS-SCCs.

### 3.5. Correlations between Immune Markers and Clinicopathological Parameters

We questioned whether the expression of IDO and PD-L1 in tumor cells correlated with the vertical tumor thickness of the EB-SCCs and thus might have prognostic significance since a high vertical tumor thickness increases the likelihood of metastatic disease [42] (Figure 5). IDO expression in tumor cells correlated well with the vertical tumor thickness in IC-SCCs (*p* < 0.004) (Figure 5). This was not seen in DEB-SCCs (*p* = 0.4047) or KEB-SCCs (*p* = 0.1194) (Appendix A). According to the PD-L1 abundance in tumor cells, we found a correlation with the tumor thickness in IC-SCCs (*p* < 0.0038) and IS-SCCs (*p* = 0.0278) (Figure 5). In EB tumors, a PD-L1 correlation was only seen in KEB-SCCs (*p* = 0.0013) and not in DEB-SCCs (*p* = 0.2125) (Appendix A). Of note, some very thin DEB-SCCs expressed high levels of IDO and PD-L1, explaining the missing correlation with vertical tumor thickness. Thus, IDO and PD-L1 expression may serve as an indirect prognostic marker, especially in IC- and IS-SCCs.

A similar result was obtained for IDO expression in TME cells, whereby a significant correlation between IDO expression and vertical tumor thickness was found in IC-SCCs (*p* = 0.0014) and IS-SCCs (*p* = 0.0229) (Figure 5). Again, no correlation was seen in DEB-SCCs (*p* = 0.6683) and KEB-SCCs (*p* = 0.6377) (Appendix A). PD-L1 expression in TME cells strongly correlated with vertical tumor thickness in IC-SCCs (*p* = 0.0022) (Figure 5). A similar trend, but no statistical significance, was obtained for PD-L1 expression in IS-SCCs (*p* = 0.1549), DEB-SCCs (*p* = 0.0637), and KEB tumor cells (*p* = 0.1475) (Figure 5 and Appendix A). Thus, the expression of IDO and PD-L1 in TME cells is not as clear as that in tumor cells. However, in IC-SCCs, the correlation with vertical tumor thickness was strong and may have prognostic value. We also analyzed the correlation with the tumor grading. The limitation was the low number of grade II and III/IV EB-SCCs and the lack of grade III/IV IC-SCCs. There was a significant correlation between the IC-SCC tumor grading and LAG-3 (*p* < 0.0163), TIM3 (*p* = 0.014), PD-L1 in the tumor (*p* < 0.0001), and PD1 (*p* = 0.029) in IC-SCCs, but none for EB-SCCs and IS-SCCs.

Next, we analyzed the expression of further immune checkpoints in the TME. LAG-3 and PD-1 expression in IC-SCCs showed a significant correlation with the Breslow thickness (*p* = 0.0163 and *p* = 0.0004, respectively) (Figure 5). No correlation of these markers with tumor thickness was found in IS-SCCs, DEB-SCCs, or KEB-SCCs (Figure 5 and Appendix A). Taken together, a high expression of checkpoint markers in T cells correlates with a high tumor thickness, especially in the TME of IC-SCCs, and may thus predict a bad prognosis.

We also analyzed whether patients in the IS-SCC group receiving combination therapy of immunosuppressive drugs showed more tumors with a high vertical tumor thickness and, thus, more aggressive behavior. We did not detect any correlation between the type of immunosuppression or the number of immunosuppressive drugs and the vertical tumor thickness.

## 4. Discussion

Understanding the underlying mechanism via which cancer cells can ‘molecularly cloak’ themselves and remain hidden from immune surveillance is of great interest to cancer biology and provides the rationale for the design of therapeutic interventions. Our study revealed the expression patterns of several immune checkpoint proteins in SCCs occurring in EB compared with immunocompetent and immunosuppressed patients. In these three groups, SCCs arose in different circumstances: (i) at a young age in damaged skin in EB; (ii) at an older age in those with an immunosuppressed background; or (iii) UV-induced in aged immunocompetent individuals. Although the number of EB-SCCs might seem low, it is significant in the context of ultra-rare disorders and compared with other studies.

Notably, we found that, in EB-SCCs, tumor cells expressed immune checkpoints that are involved in the suppression of the immune response. Chronic tissue inflammation and damage, extracellular matrix remodeling, and bacterial challenge [43,44] may contribute to checkpoint inhibitor expression to subvert the immune system early in favor of cancer development and progression. DEB-SCCs demonstrated high levels of PD-L1 and IDO expression in tumor cells and PD-1 in the TME, which was independent of tumor thickness. Similarly, KEB-SCCs showed high PD-L1 expression in tumor cells and PD-1 expression in the TME. The regulation of PD-L1 may result from transcription factors (e.g., STAT3) or cytokines (e.g., IFN, TNF, and IL-17) that are increased in DEB [45,46,47]. IDO is upregulated by tumor cells to support an immunosuppressive environment. For example, the expression levels of IDO in cutaneous melanoma cells correlate with reduced progression-free survival in these patients. Phase II trials combining PD-1 with IDO inhibitors showed promising results with objective response rates (ORR) of 51%, a complete response rate of 20%, and a disease control rate of 70% [48]. A phase I/II trial of an immune-modulatory vaccine against IDO/PD-L1 in combination with nivolumab showed an ORR of 80%, with 43% complete responses in metastatic melanoma patients [49].

In DEB-SCCs, the number of CD4^+^ cells in the TME was lower than in IC-SCCs. Our results are in line with previous reports showing a significant reduction in CD4^+^ immune cell peritumoral infiltration in DEB-SCCs compared with UV-induced SCCs, while there was no difference with IS-SCCs [50,51]. A high baseline density of TILs was associated with improved outcomes in several solid tumors treated with immune checkpoint inhibitors [52]. CD8^+^TILs are associated with prognostic benefits, whereas the role of CD4^+^TILs is controversial. In the latter, a Th1 phenotype seems to predict improved overall survival [52]. TIL activation markers and/or effector molecules allow a more accurate prediction of their prognostic value. For example, pre-existing PD-1^+^ CD8^+^TILs are associated with good treatment responses to PD-1 inhibitors in melanoma patients [53]. The prognostic role of further T cell exhaustion markers such as TIM-3 and LAG-3 is being intensively studied [52]. PD-1 and TIM-3 are frequently co-expressed during the differentiation of T cell exhaustion [54], and, in line with this, we found TIM-3 expression in TME cells. TIM-3 was similarly expressed in the SCC groups, except for in KEB-SCCs, which demonstrated significantly lower levels. This contrasts with another study showing increased TIM-3 expression in DEB-SCCs compared with primary SCCs [50]. TIM-3 is a member of the TIM family of immunoregulatory proteins and can be expressed in IFNγ-producing CD4^+^ and CD8^+^ T cells [55]. It is a marker for the most dysfunctional subset among tumor-infiltrating CD8^+^PD-1^+^ T cells in cancer [55]. A phase I/Ib trial of the anti-TIM-3 antibody alone or in combination with an anti-PD-1 antibody showed evidence for antitumor activity in advanced solid tumors [56]. We detected considerable LAG-3 expression in the TME of almost all SCCs, most prominently in IS-SCCs. Elevated LAG-3 expression has been found in advanced cutaneous SCCs, along with PD-L [57]. LAG-3 is a member of the immunoglobulin superfamily and exerts various biological impacts on T cell function [40,58]. LAG-3 binds to major histocompatibility complex-II (MHC-II) on antigen-presenting cells and is expressed on the cell membranes of TILs, leading to an exhausted phenotype [40]. Antibodies against LAG-3 have been successfully used in melanoma patients, especially in combination with anti-PD-1 antibodies [59,60].

Our results show that EB-SCCs differ in the expression of immune markers compared to IC- and IS-SCCs. DEB- and KEB-SCCs also showed differences, probably because chronic wounds do not occur in KEB and because of the complex roles of kindlin-1.

While low TIL numbers, poor differentiation, and lower TMB most likely contribute to the more aggressive behavior of EB-SCCs, the high expression of PD-L1, IDO, and PD-1 in tumor cells and their TMEs represents therapeutic targets. IDO inhibitors or IDO vaccination in combination with PD-1 inhibition may help achieve better treatment responses. TIM-3 and LAG-3 inhibitors are further combination candidates for PD-1 inhibition, and bispecific antibodies are already being tested, showing promising antitumor efficacy [61,62,63]. These results and our recent case reports [64] further the immunological characterization of advanced cutaneous SCCs in EB patients to tailor a biologically meaningful treatment.

## 5. Conclusions

The results of our study show that EB-SCCs differ in the expression profiles of immune markers and the number of TILs compared with IC- and IS-SCCs. Moreover, these results further the immunological characterization of advanced cutaneous SCCs in EB patients to tailor biologically stratified treatment decisions when therapies are needed. 

## Figures and Tables

**Figure 1 cancers-16-00471-f001:**
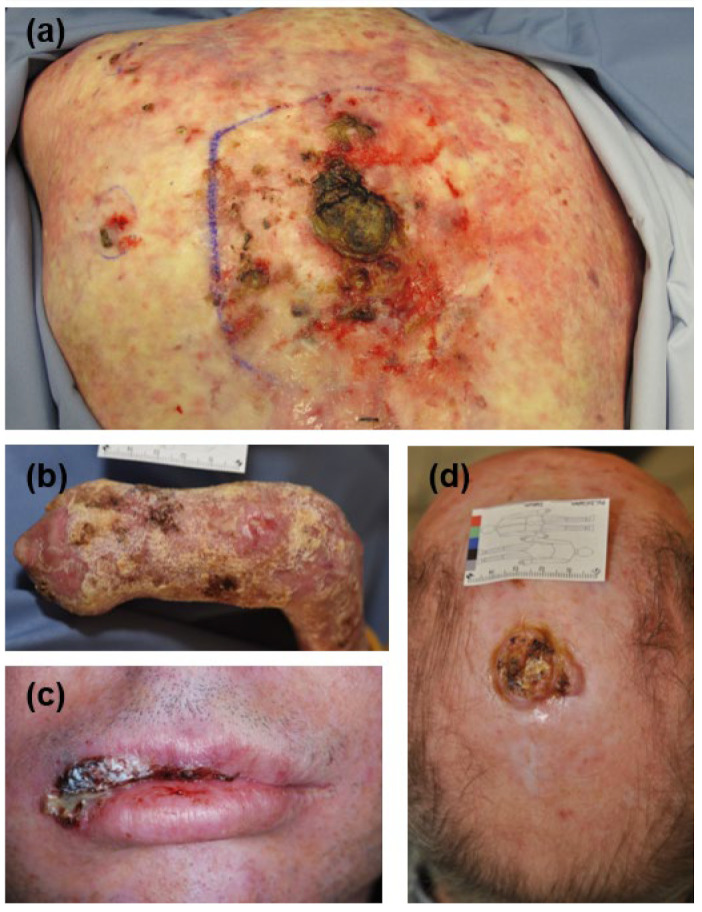
Clinical features of cutaneous SCCs in the studied groups. (**a**) SCC on the back in an organ transplant patient under double immunosuppressive therapy. Typical poikilodermic alterations of the skin because of severe actinic damage are present on the back. (**b**) Multiple SCCs on the mutilated hand of a patient suffering from dystrophic EB. (**c**) SCC in the upper lip in a patient with Kindler EB. (**d**) A hyperkeratotic SCC on the scalp in an immunocompetent patient. EB, epidermolysis bullosa; SCC, squamous cell carcinoma.

**Figure 2 cancers-16-00471-f002:**
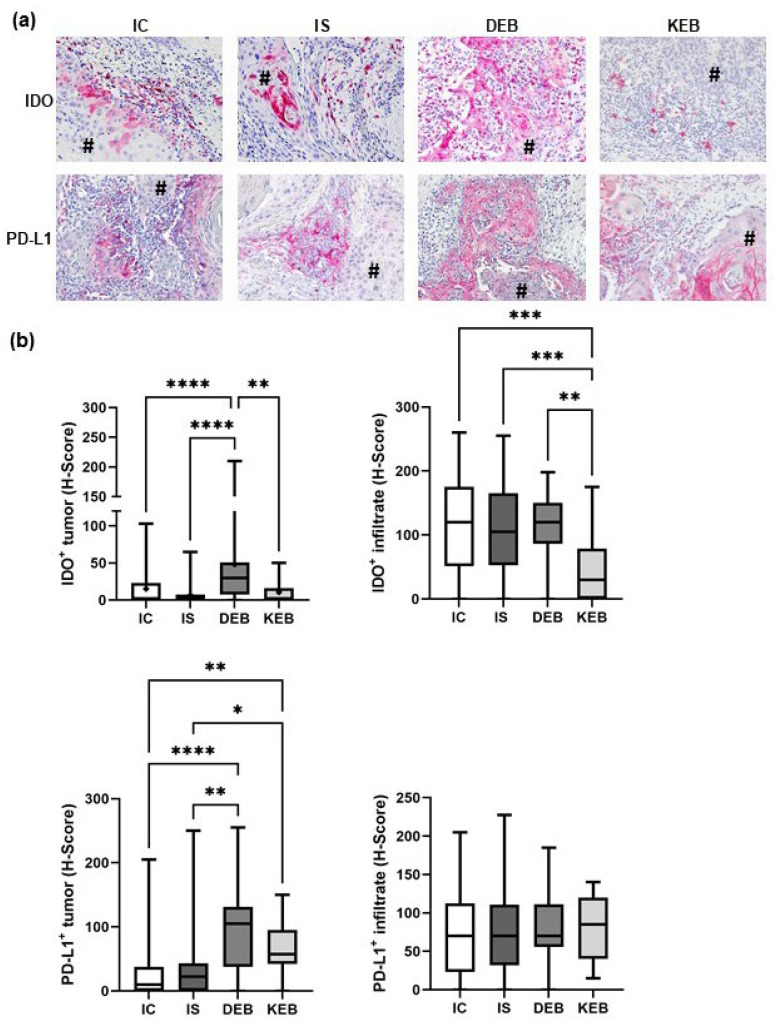
IDO and PD-L1 expression in cutaneous SCCs and their TMEs. (**a**) Representative pictures of immunohistochemical (IHC) stainings from each SCC subgroup showing IDO and PD-L1 expression in cells of the tumors and of their microenvironments. # marks tumor cells. Magnification = 200×. (**b**) Statistical analysis comparing mean expression levels of IDO and PD-L1 in the SCC groups. Staining intensities were quantified using the H-scores. IC = immunocompetent, IS = immunosuppressed, DEB = dystrophic EB, and KEB = Kindler EB. Statistical significance was considered as * *p* < 0.05; ** *p* < 0.01; *** *p* < 0.001; **** *p* < 0.0001.

**Figure 3 cancers-16-00471-f003:**
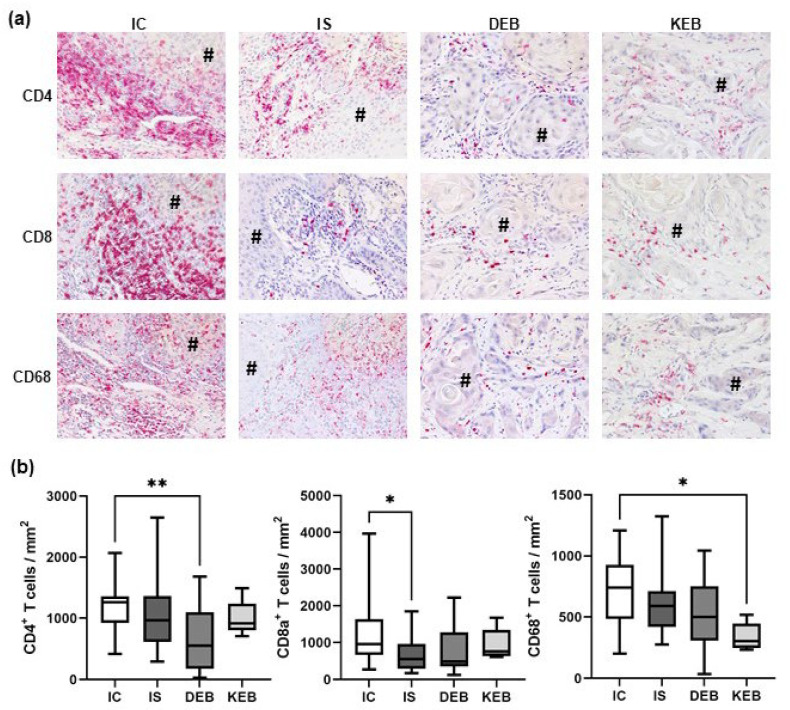
Inflammatory infiltrates in the TMEs of cutaneous SCCs. (**a**) Representative pictures of immunohistochemical stainings from each SCC group showing CD4^+^ and CD8^+^ T cell infiltrates as well as CD68^+^ macrophages in the tumor microenvironment. # marks tumor cells. Magnification = 200×. (**b**) Statistical analysis comparing mean absolute cell numbers per mm^2^ of CD4^+^, CD8^+^, and CD68^+^ cells in all SCC groups. IC = immunocompetent, IS = immunosuppressed, DEB = dystrophic EB, and KEB = Kindler EB. Statistical significance was considered as * *p* < 0.05 and ** *p* < 0.01.

**Figure 4 cancers-16-00471-f004:**
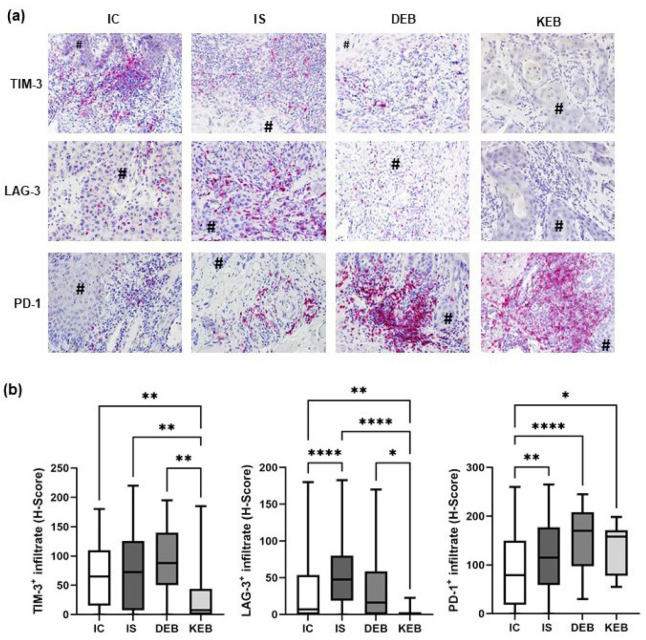
TIM-3, LAG-3, and PD-1 expression in the TME of cutaneous SCCs. (**a**) Representative pictures of immunohistochemical stainings from each SCC group showing TIM-3, LAG-3, and PD-1 expression in TME stromal cells. # marks tumor cells. Magnification = 200×. (**b**) Statistical analysis comparing mean expression levels of TIM-3, LAG-3, and PD-1 in all SCC groups. Staining intensities were quantified using the H-scores. IC = immunocompetent, IS = immunosuppressed, DEB = dystrophic EB, and KEB = Kindler EB. Statistical significance was considered as * *p* < 0.05, ** *p* < 0.01, and **** *p* < 0.0001.

**Figure 5 cancers-16-00471-f005:**
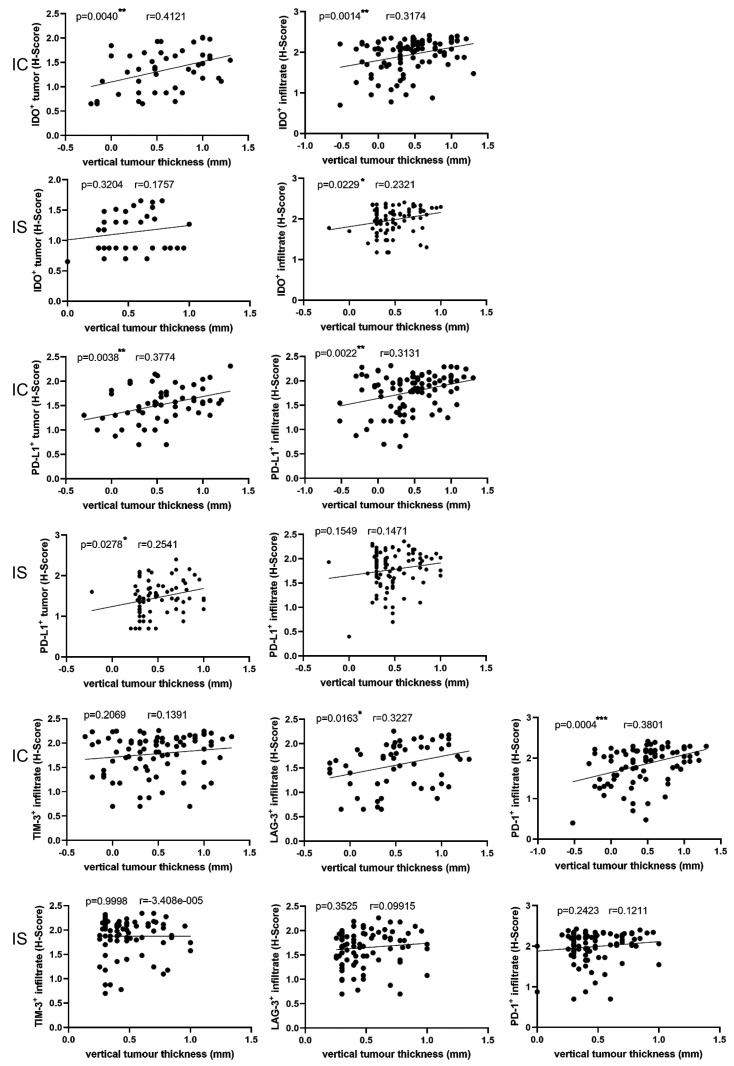
Correlation of immune marker expression with vertical tumor thickness. Statistical analysis showed how the expression of the markers (*y*-axis) in tumor cells and stromal cells is affected by vertical tumor thickness (*x*-axis) using a simple linear regression model. IC = immunocompetent, IS = immunosuppressed. Statistical significance is noted as * *p* < 0.05, ** *p* < 0.01, *** *p* < 0.001. r = Pearson correlation coefficient. Measured values of *x*- and *y*-axis were logarithmized before statistical analysis.

**Table 1 cancers-16-00471-t001:** Clinical and pathological characteristics of the patients and SCCs.

	DEB ^1^-SCC ^2^	KEB ^3^-SCC	IC ^4^-SCC	IS ^5^-SCC
**Number of patients/SCC samples**	9/30	7/22	96/100	41/106
**Mean age at SCC** **diagnosis (range) in years**	32 (18–50)	47 (30–65)	80 (45–99)	68 (41–87)
**Gender (male:female)**	4:5	5:2	58:38	27:14
**Localization—number (%)**				
Head and neck	0 (0)	7 (32)	79 (79)	54 (51)
Trunk	0 (0)	0 (0)	0 (0)	17 (16)
Upper extremities	16 (53)	5 (23)	19 (19)	24 (23)
Lower extremities	14 (47)	10 (45)	2 (2)	11 (10)
**Histologic grading—number (%)**				
G1	22 (73)	8 (36)	48 (48)	38 (36)
G2	5 (17)	3 (14)	51 (51)	60 (57)
G3	3 (10)	11 (50)	1 (1)	8 (7)
**Vertical tumor thickness in mm (%)**				
≤2	7 (23)	6 (27)	42 (42)	35 (33)
2.01–6	14 (47)	16 (73)	40 (40)	58 (55)
>6	5 (17)	0 (0)	18 (18)	9 (8)
NA ^6^	4 (13)	0 (0)	0 (0)	4 (4)

^1^ Recessive dystrophic EB; ^2^ squamous cell carcinoma; ^3^ Kindler EB; ^4^ immunocompetent; ^5^ immunosuppressed; ^6^ not available.

## Data Availability

Data are contained within the article and Appendix A.

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
