# Peer review of "Epidermolysis-Bullosa-Associated Squamous Cell Carcinomas Support an Immunosuppressive Tumor Microenvironment: Prospects for Immunotherapy"

_cancers, 2024, doi:10.3390/cancers16020471_

Round 1

Reviewer 1 Report

Comments and Suggestions for Authors

Authors present an immunohistochemical study of skin SCC from immunocompromised patients, immunocompetent patients and patients with two forms of epidermolysis bullosa (EB), dystrophic and kindler EB. Authors report data on a good number of samples in each cohort for markers identifying CD8, CD4 and CD68 positive immune cells as well as the immune checkpoint related markers IDO, PD-L1, TIM-3, LAG-3 and PD-1.

Overall the study is sound and reports important data in the context of EB cancer. Some issues are present with respect to interpretation of the data and how the findings are discussed.

Authors should report on the data and not their own conjecture with respect to interpretation. The title should be improved and focus on the data reported. Since reports of responses to immune checkpoint inhibition (ICI) in EB cancer have been variable and it is widely known that while some cancers in other contexts respond well to ICI there has been little agreement on markers that predict response as many tumors with features that would suggest response (such as high PD-1/PD-L1) in fact do not respond. Equally while the data support the idea that the TME in EB is immunosuppressive they do not demonstrate this.

Specific instances which should be re-worded:

Simple Summary "presumably turning these tumors susceptible". Please avoid presumption in summaries/ abstracts and results.

Last sentence of introduction "Here, we investigate whether EB-SCC develop such resistant mechanisms..." should be changed since no temporal analaysis has been carried to comment on "develop" and no mechanistic studies are being undertaken.

Page 7, line 212, use of "poor" in the title is conjecture without qualifying what "poor" refers to. Please stick to reporting the data in the results section. What constitutes a "representative number"?

Page 8, line 233 and 249. The use of the verb "Dominate" is not qualified by analyzing 3 proteins nor do the numbers seem to support domination.

Page 9, line 260, "Predictive value of immune markers..." is misleading since there isn't application of data reported to cohorts of patients to assess whether a prediction can be made on the few protein markers analyzed. Please report data factually in the results section.

Page 9, line 288, introduction and single reference to Breslow at this point is confusing. Isn't it already known whether tumor thickness is predictive of prognosis in multiple other studies, why start predicting this here?

Discussion, page 10, line 308, "Our study reveals distinct patterns of immune evasion" is overstating the data since the study really doesn't do this. Please make discussive statements in the context of the data and be mindful of the limitations of studying a handful of IHC markers when doing so.

Page 11, line 315 "we found that in EB-SCC, tumor cells actively suppress the immune response" is an overstatement since the study is purely observational and does not investigate EB-SCC tumor cells and their ability to suppress immune response.

Page 11, line 323, "Our results... ...suggest early treatment". Again no temporal data is presented so timing can not be inferred.

Page 11, line 333, only CD4+ were significantly lower.

Line 362 - define "distinct profiles"

Information on tumor size (area, regardless of tumor thickness), if available, would be important to report or analyze.

Some issues with consistency are evident throughout, data are called out differently in the simple summary compared with the abstract (no mention of PD-1 in summary).

Some typogrpahical/ grammatical issues, see introduction, second sentence, add EB after DEB, subtypes after (KEB) and "a" to "occur at "a" young age".

Author Response

Answers to the comments of Reviewer 1

We thank the reviewer for carefully assessing our manuscript. 

  1. Simple Summary "presumably turning these tumors susceptible". Please avoid presumption in summaries/ abstracts and results.

Answer: We changed the text to read:

“Our results show high expression of the immunosuppressive markers indoleamine 2,3-dioxygenase (IDO), PD-1 and programmed cell death ligand-1 (PD-L1) in tumor cells from dystrophic EB (DEB) patients.”

“These findings identify IDO, PD-1 and PD-L1 to be increased in EB-SCC and candidate targets for combinatory treatments, especially in DEB-SCC.”

  1. Last sentence of introduction "Here, we investigate whether EB-SCC develop such resistant mechanisms..." should be changed since no temporal analysis has been carried to comment on "develop" and no mechanistic studies are being undertaken.

Answer: The sentence was changed to read

“This study aims to investigate whether EB-SCC express immune checkpoint markers that might contribute to the severe course and might be therapeutically targeted.”

  1. Page 7, line 212, use of "poor" in the title is conjecture without qualifying what "poor" refers to. Please stick to reporting the data in the results section. What constitutes a "representative number"?

Answer: This title and the text were changed:

“DEB-SCC tumor cells express significantly higher levels of IDO and PD-L1 compared to IC- and IS-SCC”

“Next, we quantified inflammatory cells, CD4+ and CD8+ T cells, and CD68+ macro-phages in the TME of 25-30% of the SCC of each group.”

  1. Page 8, line 233 and 249. The use of the verb "Dominate" is not qualified by analyzing 3 proteins nor do the numbers seem to support domination.

Answer: The word dominate was eliminated.

  1. Page 9, line 260, "Predictive value of immune markers..." is misleading since there isn't application of data reported to cohorts of patients to assess whether a prediction can be made on the few protein markers analyzed. Please report data factually in the results section.

Answer: This title was changed:

“Correlations between immune markers and clinicopathological parameters”

  1. Page 9, line 288, introduction and single reference to Breslow at this point is confusing. Isn't it already known whether tumor thickness is predictive of prognosis in multiple other studies, why start predicting this here?

Answer: We changed the sentence to read:

“To this end, we asked whether the expression of IDO and PD-L1 in/on tumor cells correlates with the vertical tumor thickness of the EB-SCC and thus might have a prognostic significance since high vertical tumor thickness is known to increase the likelihood of metastatic disease”

  1. Discussion, page 10, line 308, "Our study reveals distinct patterns of immune evasion" is overstating the data since the study really doesn't do this. Please make discussive statements in the context of the data and be mindful of the limitations of studying a handful of IHC markers when doing so.

Answer: We changed the sentence to read:

“Our study reveals the expression patterns of several immune checkpoint proteins in SCC occurring in EB, as compared to immunocompetent and immunosuppressed patients.”

  1. Page 11, line 315 "we found that in EB-SCC, tumor cells actively suppress the immune response" is an overstatement since the study is purely observational and does not investigate EB-SCC tumor cells and their ability to suppress immune response.

Answer: We changed the sentence to read:

            “we found that in EB-SCC, tumor cells express immune checkpoints that are       involved in the suppression of the immune response.”

  1. Page 11, line 323, "Our results... ...suggest early treatment". Again no temporal data is presented so timing can not be inferred.

            Answer: This was deleted.

  1. Page 11, line 333, only CD4+ were significantly lower.

Answer: We eliminated CD8+. These were also reduced but statistical significance was not achieved.

  1. Line 362 - define "distinct profiles"

Answer: We changed the text to read:

“Our results show that EB-SCC differ in the expression of immune markers from IC- and IS-SCC. DEB- and KEB-SCC also showed differences,”

  1. Information on tumor size (area, regardless of tumor thickness), if available, would be important to report or analyze.

Answer: This information is available only for a small number of tumors. Therefore we could not use it for analysis.

  1. Some issues with consistency are evident throughout, data are called out differently in the simple summary compared with the abstract (no mention of PD-1 in summary).

Some typogrpahical/ grammatical issues, see introduction, second sentence, add EB after DEB, subtypes after (KEB) and "a" to "occur at "a" young age".

Answer: We corrected typos and grammatical issues. Because of the short revision time, professional correction was not possible.

Reviewer 2 Report

Comments and Suggestions for Authors

The authors aimed to evaluate the immune microenvironment of cutaneous SCC that is assumed to favor local immunosuppression and lead to a more severe disease course in these patients. The expression of several immune checkpoint molecules in tumor cells and cells of the tumor microenvironment was evaluated in EB-SCC and compared to that in SCC from immunocompetent individuals and from immunosuppressed patients. Their results show high expression of the immunosuppressive markers indoleamine 2,3-diox-ygenase (IDO) and programmed cell death ligand-1 (PD-L1) in tumor cells from dystrophic EB (DEB) patients presumably turning these tumors susceptible to therapies which specifically target these molecules.

The study, is easy to follow and covers an hot topic, but some issues should be improved before publication.

The manuscript needs moderate English change and grammar correction.

Several typos should be corrected thorough the text.

Itroduction Section: the authors should better explain the aim of the study.

Conclusion Section: This paragraph required a general revision to eliminate redundant sentences and to add some "take-home message".

Comments on the Quality of English Language

Minor editing of English language required

Author Response

Answers to Reviewer 2

We thank the reviewer for comments.

The study, is easy to follow and covers an hot topic, but some issues should be improved before publication.

Answer: We thank the reviewer for this comment.

The manuscript needs moderate English change and grammar correction.

Several typos should be corrected thorough the text.

Answer: We corrected typos and grammatical issues. Because of the short revision time, professional correction was not possible.

Introduction Section: the authors should better explain the aim of the study.

Answer: This study aimed to investigate whether EB-SCC express immune checkpoint mark-ers that might contribute to the severe course and might be therapeutically targeted.

Conclusion Section: This paragraph required a general revision to eliminate redundant sentences and to add some "take-home message".

Answer: We changed the conclusion to read:

The results of our study show that EB-SCC differ in the expression profile of immune markers and the number of TILs compared to IC- and IS-SCC and advocate the immuno-logical characterization of advanced cutaneous SCC in EB patients to tailor a biologically stratified treatment decision when therapies are needed.

Reviewer 3 Report

Comments and Suggestions for Authors

Statistics:

You performed a lot of statistical tests. Please consider correcting the p-values for multiple testing.

Figures 2, 3 and 4: the statistical graphs are hard to read. Please remove all indications of “ns”. In addition, please either use simple box plots or violin charts instead of depicting all data points. This makes the interpretation very difficult.

In Figure 5, one can see that most data points are concentrated on lower values, therefore making an interpretation of the correlation very difficult. I advise trying to logarithm the values before performing linear regression and correlation analysis. Another way out would be to use a logarithmic scale. Anyway, you cannot leave it the way it is.

Author Response

Reviewer 3

You performed a lot of statistical tests. Please consider correcting the p-values for multiple testing.

Answer:

“Kruskal-Wallis test and Dunn’s multiple comparison tests were used to compare SCC groups.”

Figures 2, 3 and 4: the statistical graphs are hard to read. Please remove all indications of “ns”. In addition, please either use simple box plots or violin charts instead of depicting all data points. This makes the interpretation very difficult.

Answer: The graphs were changed into box plots.

In Figure 5, one can see that most data points are concentrated on lower values, therefore making an interpretation of the correlation very difficult. I advise trying to logarithm the values before performing linear regression and correlation analysis. Another way out would be to use a logarithmic scale. Anyway, you cannot leave it the way it is.

Answer: The logarithmic representation did not help the interpretation of the results in our hands. We would like to maintain the graphs as they are.

Round 2

Reviewer 3 Report

Comments and Suggestions for Authors

Thank you very much for improving the figures! Still, I would like to see a comparison of Figure 5 to a potential new figure, in which the measured values were logarithmised. Probably, your p-values will even improve.

In addition, in Figure 5 and the corresponding text, you only show the p-values of the correlation. Where are the corresponding correlation coefficients? Please add those. You could compare the correlation coefficients between the analyses before logarithmisation and afterwards and then let me know.

Author Response

We provide the revised figure 5 as compared to the initial and kindly ask the reviewer and editor for advise.

Round 3

Reviewer 3 Report

Comments and Suggestions for Authors

The authors added R-squared to Figure 5, which is the square of the correlation coefficient. (https://en.wikipedia.org/wiki/Coefficient_of_determination)

However, they indicated the p-values of the correlation analyses. These two parameters do not match.

They need to add the Pearson Correlation coefficient:

https://en.wikipedia.org/wiki/Pearson_correlation_coefficient

In addition, I asked them TWICE to use the logarithm of their measured values and then repeat Figure 5. They did not do this, but instead, they just altered a little bit the x-axis. This is not the same.

https://en.wikipedia.org/wiki/Logarithm

I would recommend again to compare original Figure 5 (with correct correlation coefficients) to a new figure 5 with the logarithm applied to the measured values.

Author Response

Point by point response

Reviewer 3
The authors added R-squared to Figure 5, which is the square of the
correlation coefficient.
(https://en.wikipedia.org/wiki/Coefficient_of_determination)
However, they indicated the p-values of the correlation analyses.
These two parameters do not match.
They need to add the Pearson Correlation coefficient:
https://en.wikipedia.org/wiki/Pearson_correlation_coefficient

Answer: We added the Pearson Correlation coefficient to the
analysis.

In addition, I asked them TWICE to use the logarithm of their
measured values and then repeat Figure 5. They did not do this, but
instead, they just altered a little bit the x-axis. This is not the same.
https://en.wikipedia.org/wiki/Logarithm
I would recommend again to compare original Figure 5 (with correct
correlation coefficients) to a new figure 5 with the logarithm applied
to the measured values.

Answer: We apologize for the misunderstanding. We now provide an
alternative figure 5 (version 2) in which the values of the X- and the
Y-axis have been logarithmized before statistical analysis. The new pvalues
and Pearson Correlation coefficients are displayed accordingly.

Best regards,

David Rafei and Cristina Has.

Round 4

Reviewer 3 Report

Comments and Suggestions for Authors

Thank you very much; everything is fine now!